# Distribution, Ecological Risk Assessment, and Bioavailability of Cadmium in Soil from Nansha, Pearl River Delta, China

**DOI:** 10.3390/ijerph16193637

**Published:** 2019-09-27

**Authors:** Fangting Wang, Changsheng Huang, Zhihua Chen, Ke Bao

**Affiliations:** 1School of Environmental Studies, China University of Geosciences, Wuhan 430074, China; 2Wuhan Geological Survey Center, China Geological Survey, Wuhan 430205, China; 3Changjiang Water Resources Commission of the Ministry of Water Resources, Wuhan 430010, China

**Keywords:** heavy metal, spatial distribution, ecological risk assessment, cadmium bioavailability

## Abstract

*Background:* Cadmium (Cd) pollution poses a threat to human health. Examination of the spatial distribution of Cd in soils can be used to assess the risks posed to humans and the environment. *Objective:* This study determined the enrichment rules and factors influencing Cd pollution in Nansha, and evaluated the pollution characteristics and bioavailability of Cd in quaternary sediments through 7 deep soil profiles (0–200 cm), 4 boreholes, and 348 topsoil (0–20 cm) samples. *Methods:* The geo-accumulation index (Igeo) and the potential ecological risk index (Er) were used to assess ecological risk, and bioavailability was determined using multivariate, spatial distribution, and correlation matrix analyses. *Results:* From the Er, 52% of Nansha was classed as being at very high risk of Cd pollution; a further 36% was classed as dangerous. Cadmium was more abundant in clay soils than in sandy soils. Bioavailable Cd in quaternary sediments was significantly affected by the total Cd, and labile Cd accounted for more than half of the total Cd. Changes in pH mainly affected bioavailable Cd rather than total Cd, affecting the overall bioavailability of Cd. *Conclusions:* Nansha soils are commonly and seriously contaminated with Cd. An appropriate remediation treatment approach should be used to reduce Cd bioavailability. Furthermore, planting structures in farmland should be adjusted to avoid the impact of heavy metals on human health.

## 1. Introduction

Rapid urbanization is a worldwide phenomenon accompanied by intensive industrial and economic activities and environmental problems, especially in developing countries [1]. Industrialization/urbanization (e.g., industry, agriculture, and transportation) and natural geological processes can lead to the heavy metal pollution of some farmland soils [2,3]. Heavy metals in soils pose a threat to human health through the food chain [4,5], and cadmium (Cd) is an important pollutant among various heavy metal elements, due to its high migration rate from soil to plants and strong biological toxicity [6]. There is increasing evidence that environmental exposure to Cd is associated with increased cancer incidence [7]. The excessive input of heavy metals into soils has attracted extensive worldwide attention due to their toxicity, durability, and biological enrichment [8], and heavy metal pollution in soil has become a globally recognized environmental problem [9]. The soil pollution situation is a particular problem in China. The development of industry and agriculture has meant that Cd in soil has significantly increased due to the extensive application of fertilizers and pesticides, the agricultural utilization of industrial wastewater and sludge, and the increasing atmospheric deposition of heavy metals. Cadmium pollutants were found to exceed the standard at 7.0% of the survey points within 6.3 million square kilometers, according to the National Soil Pollution Condition Investigation Gazette (2014). The area of Cd-contaminated farmland is 2800 km^2^, and some sewage-irrigated areas are used to produce “Cd rice” [10], which generally refers to rice of which the cadmium content exceeds the standard. The safety standard for cadmium in rice in China is 0.2 mg/kg, The annual output of Cd-contaminated agricultural products exceeds 1.5 million tons [11].

In recent years, researchers have paid more attention to the spatial distribution of heavy metals in soils in order to assess the potential risks to humans and the environment [6,12,13,14]. The spatial distribution of pollutants has been used to quantify the level of pollution in an area, and the identification of influencing factors can be used to measure the effectiveness of pollution abatement measures [15]. Previous studies have investigated the effect of soil parent material [16,17], soil type and physicochemical properties [18,19,20,21], the water environment [22], atmospheric deposition [23,24], and human activities (traffic emissions, industrial emissions, etc.) [25,26] on soil Cd pollution. However, the impact of these different factors is often regional. Therefore, it is necessary to conduct a more in-depth analysis of the degree and scope of the influence of these factors on soil Cd pollution. Although the total heavy metal levels in a soil can reflect the degree of enrichment by heavy metals in soils, the bioaccessibility and biotoxicity of heavy metals are more dependent on their fractionation bioavailability [27], which has gradually become an important basis for soil pollution assessment and risk prediction [17].

This study took the Nansha District of the Pearl River Delta, China, as the research area. Field sampling and laboratory sample analyses were undertaken, with the objectives of (1) quantifying and determining the spatial distribution patterns of soil Cd in the Nansha district; (2) assessing Cd pollution levels and potential ecological risks; and (3) quantifying the bioavailability of Cd and identifying the factors affecting Cd bioavailability in the soil. The data obtained in this study will provide a basis for the prevention, treatment, and remediation of soil Cd pollution in farmland around cities.

## 2. Materials and Methods

### 2.1. Study Area

Nansha District is a municipal district of Guangzhou, Guangdong Province, which became the sixth new national area in October, 2012. It is located at the southernmost end of Guangzhou city on the west bank of the Humen waterway, which is part of the Pearl River. It is where the Xijiang River, Beijiang River, and Dongjiang River converge, and is in the geometric center of the Pearl River Delta, which is the channel from the Pearl River basin to the ocean. The landforms in Nansha are hilly, marine–terrigenous plains, and tidal flats. The low hills are mainly distributed around Huangshanlu Mountain and the highest point in the region is 295.3 m above sea level. The marine–terrigenous plains are distributed in Huangge and Hengli; the upper part is silt or silty soil (sand) and the lower part is river-deposited sand. The tidal flats are mainly distributed in the southeastern area of Wanqingsha, Longxue Island, and the Xinken coast. The terrain is zonal, parallel to the coast, broad, and gentle. In March, 2017, the development plan for the Guangdong–Hong Kong–Macao greater bay area was proposed. Nansha, as the geometric center of the greater bay area, is also the pivot for Guangzhou’s development of the “One Belt One Road” initiative, and the maritime Silk Road forms the Guangdong–Hong Kong–Macao international strategic platform for the greater bay area. 

### 2.2. Sampling

The topsoil (0–20 cm) sampling points were based on the 1:50,000 standard map, and the density was generally four samples per km^2^. A total of 348 samples were collected, based on geological background, land use, and the topographical and geomorphic characteristics of the sampling cells (Figure 1). During sampling, wooden sampling tools were used to collect 0–20 cm depth cubic soil columns, and the weeds, grass roots, gravel, fertilizer clumps, and other sundries in the samples were removed. One sample was taken as a fixed point and combined with 3–5 sub-samples collected within a radius of 50 m to increase the representativeness of the soil samples. The samples were placed in new, sealed bags.

Seven deep soil (0–200 cm) profiles were arranged according to the soil genetic type, including that of residual sediments and alluvium. A hand drill was used to collect 0–2 m sections, and the soil in contact with the sampling tool was removed. Each section was divided into five samples (0–20 cm, 20–50 cm, 50–100 cm, 100–150 cm, and 150–200 cm), which were placed into new, sealed bags.

Four boreholes from the land to the sea, which exposed the quaternary sediments, were selected as sampling points, numbered NSGC27, NSGC05, NSGC11, and NSGC39. The sampling depth of the boreholes was 48.3 m, 32.5 m, 43.7 m, and 37.8 m, respectively, until fully weathered bedrock was reached. Stratified sampling was adopted to stratify the section according to the sedimentary material or lithology, and equally spaced samples were taken according to the thickness of each layer.

### 2.3. Laboratory Analysis

The samples were analyzed in the laboratory for soil acidity, cation exchange capacity (CEC), soil organic matter (SOM), potassium (K), calcium (Ca), sodium (Na), magnesium (Mg), aluminum (Al), copper (Cu), arsenic (As), mercury (Hg), zinc (Zn), iron (Fe), manganese (Mn), titanium (Ti), lead (Pb), cadmium (Cd), silicon dioxide (SiO_2_), boron (B), barium (Ba), strontium (Sr), and the contents of the seven morphological forms of Cd.

In accordance with Determination of pH Value in Forest Soil (LY/t1239-1999), soil acidity was determined using the ion-selective electrode method/pH meter, CEC was determined by the ammonium acetate exchange method, and SOM was determined by the potassium dichromate volume method. Potassium, calcium, sodium, magnesium, aluminum, copper, zinc, iron, and manganese were determined by inductively coupled plasma emission spectrometry/thermoelectric ICAP6300 in accordance with Analysis Methods for Regional (DZ/t0279.2-2016); arsenic and mercury were determined by atomic fluorescence spectrometry in accordance with Soil Testing (NY/t1121.10-2006); lead and Cd were determined by inductively coupled plasma mass spectrometry/thermoelectric X2 according to Analysis Methods for Regional (DZ/t0279.3-2016); titanium, lead, silicon dioxide, barium, and strontium were determined by powder compaction–X-ray fluorescence spectrometry in the Netherlands in accordance with Analysis Methods for Regional (DZ/t0279.1-2016); and boron was determined by AC arc emission spectrometry in accordance with Analysis Methods for Regional (DZ/t0279.1-2016). The seven-step sequential extraction method recommended by the Technical Requirements for Analysis of Ecological Geochemical Evaluation Samples (DD2005-03) was used to analyze the seven forms of Cd, namely water-soluble, exchangeable, carbonate-bound, humic-acid-bound, Fe–Mn-oxide-bound, refractory-organic-matter-bound, and residual fractions. The details of each step of the protocol are given in Table 1.

### 2.4. Soil Pollution Evaluation

#### 2.4.1. Geo-Accumulation Index

The geo-accumulation index (Igeo) was originally proposed by Muller (1969) [28] to evaluate the degree of metal pollution in sediments. It is widely used to study the degree of metal enrichment in soil, sediments, and dust. It is defined by the following equation:(1)$$I_{geo}=log_{2}\frac{C_{n}}{kB_{n}}$$
where C_n_ represents the measured concentration of metal n in the soils and B_n_ represents the geochemical background concentration of metal n in the soils. The constant k (k = 1.5) is a correction coefficient that helps to account for natural fluctuations and anthropogenic influences. The Igeo values are classified as follows: uncontaminated (Igeo ≤ 0), uncontaminated to moderately contaminated (0 < Igeo ≤ 1), moderately contaminated (1 < Igeo ≤ 2), moderately to heavily contaminated (2 ≤ Igeo < 3), heavily contaminated (3 ≤ Igeo < 4), heavily to extremely contaminated (4 ≤ Igeo < 5), and extremely contaminated (Igeo ≥ 5).

#### 2.4.2. Potential Ecological Risk Index (Er)

The potential ecological risk index (Er) was originally proposed by Hakanson (1980) [29] and is based on sedimentological theory. It not only considers the enrichment level of each element, but also their unique toxicities and comprehensive ecological risk [30]. The Er is calculated as follows [31]:(2)$$E_{r}=\frac{C_{i}}{B_{i}}T_{0}$$
where T_0_ is the heavy metal toxic response factor. This value is set at 30 for Cd (unitless) [32]. Ci and Bi (mg/kg) represent the Cd concentration in the soil samples and the soil Cd background concentration, respectively. The Er is divided into five classifications: low risk (Er < 40); moderate risk (40 ≤ Er < 80); considerable risk (80 ≤ Er < 160); very high risk (160 ≤ Er < 320); and dangerous (Er ≥ 320).

### 2.5. Multivariate Statistics and Spatial Analysis Methods

Descriptive statistics were produced and a correlation analysis was conducted using Microsoft Office documents Excel 2010 (Redmond, WA, USA). The coefficient of variation (CV) is the ratio of standard deviation (SD) to the mean, which reflects the average variation degree of a certain attribute at each sampling point. It is generally considered that the total sample has low spatial variability if CV ≤ 10%, that the spatial variability is moderate if 10% < CV < 100%, and that the spatial variability is strong if CV ≥ 100% [33]. A Pearson’s correlation matrix (PCM) [34] was prepared, and a hierarchical cluster analysis (HCA) [35] and a principal component analysis (PCA) [36] were performed using IBM SPSS statistics for Windows, version 20.0 (Chicago, Ill, USA) to quantify the correlations between each element. MAPGIS 6.7 software (Wuhan, China) was used to establish the Geosoft binary grid (GRD) mathematical model using the Kring pan-kriging method to construct the discrete data grid, and high level smooth contour line processing was adopted to draw the contour map so that the spatial distribution pattern of the studied metals could be displayed.

## 3. Results and Discussion

### 3.1. Spatial Distribution of Cd in Soil

#### 3.1.1. Vertical Distribution of Cd

##### Vertical Distribution of Cd in Deep Soil (0–200 cm)

The Cd, pH, CEC, and SOM contents in the residual slope sediments were 0.06 to 0.17 mg/kg, 4.95 to 5.21, 3.49 to 5.64 cmol/kg, and 0.11 to 0.98%, respectively, which meant that they were far lower than in the alluvium. The alluvium analysis results are shown for comparison (Figure 2). The overall trend for Cd was as follows: the deeper the soil, the lower the Cd concentration, which was consistent with the gradual decrease in CEC and SOM, but was the reverse of the trend for pH. Due to the large rainfall and the effects of human engineering activity, the CV for Cd in the topsoil was greater than in the deep soil, and the Cd levels in the different profiles at the same burial depth tended to be consistent, especially at 20–50 cm and 150–200 cm, which was consistent with the change trends for pH and SOM. NS113PM was the profile in which the Cd content decreased most significantly as soil depth increased. The lowest Cd contents were found at 150–200 cm deep for all soil profiles, which corresponded to the generally low CEC, low SOM, and generally high pH values.

##### Vertical Distribution of Cd in Quaternary Sediments

The Cd concentration in NSGC27 ranged from 55 to 388 μg/kg, NSGC05 ranged from 28 to 560 μg/kg, NSGC11 ranged from 54 to 664 μg/kg, and NSGC39 ranged from 29 to 573 μg/kg (Table 2). The Cd was more significantly enriched than the crustal abundance. The total Cd in the soils of the four drilling profiles was mostly higher than the background concentration for Guangdong Province soil of 56μg/kg. The mean and median concentrations for each drilling were significantly higher than the background concentrations for Chinese soils, continental crust abundance, and the Pearl River sediment of 97, 80, and 90μg/kg respectively. In general, the Cd concentration tended to decrease as the depth increased (Figure 3). The Cd in the soils from cores with a depth of 0–10 m was generally higher than 300 μg/kg, and the highest content reached 664 μg/kg. In addition, the Cd concentration was low in coarse sediment dominated by sandy soil, whereas it was high in fine sediment dominated by cohesive soil.

#### 3.1.2. The Plane Distribution Pattern for Cd in Topsoil (0–20 cm deep)

The pH in the topsoil from Nansha was 4.05 to 8.43. It was generally neutral to acidic, but in some local areas it was weakly alkaline. The pH was higher on both sides of the Jiaomen River and near the Pearl River estuary, and lower in Hengli, Wanqingsha, and Nansha (Figure 4). The pH of the red soil area was generally lower than that of other soil types due to the overlying slope deposit of fully weathered granite. The lowest pH for the red soil was 4.05, the mean was 5.89, and the median was 5.44 (Table 3). The CECs for Hengli, Wanqingsha, and Xinkeng, in the western part of the Jiaomen River, were generally higher than in the east, and the alluvial soil, paddy soil, and swamp soil were about 2–4 times higher than the other soil types. The median and mean contents of the organic matter were 1.82% and 1.94%, respectively, and the closer they were to the Pearl River estuary, the lower the organic matter content was.

The Cd in the Nansha topsoil was 0.01–2.68 mg/kg, and the average and median values were 0.54 mg/kg and 0.57 mg/kg, respectively, which exceeded the risk screening value of the “Soil Environmental Quality Risk Control Standard for Soil Contamination of Agricultural Land (GB15618-2018)” of 0.3mg/kg. The soil Cd varied within and between towns, and the soil Cd in Hengli and Xinken farmland soils exceeded 0.6mg/kg. The contour map for topsoil Cd concentration (Figure 5) shows that the soil Cd distribution in the Nansha district was regular, and that (1) the soil Cd concentrations in Hengli town, Wanqingsha town, and Xinken town on the southwest bank of the Jiaomen waterway were generally higher than in Nansha on the northeastern bank of the Jiaomen waterway. These differences are related to the landforms and stratum lithologies. The southwest bank of the Jiaomen waterway is a delta alluvial plain, with flat and open terrain and a relative height difference of less than 2 m. The soils belong to the sea–land sea alluvial lantern group ($Q_{4dl}^{3mc}$). The northeastern bank of the Jiaomen waterway is dominated by a hilly landform and the quaternary strata are alluvial residual soils ($Q^{el}$); (2) there were many Cd extremities in the soils along both sides of the waterway; (3) the adsorption–desorption of Cd by soil is affected by soil type, soil solution, and soil chemical, and mineralogical characteristics, including pH, SOM, CEC, iron and manganese oxide levels, etc. [37]. However, the Cd in the soils from the northeastern coast of the Jiamen waterway was generally low, whereas the SOM level was generally high. The Cd content had a very weak correlation with pH, SOM, and CEC, and the correlation coefficients were 0.25, 0.159, and 0.171, respectively; and (4) the Cd content varied with soil type. The Cd content order was coastal sand > swamp soil > paddy soil > fluvo-aquic soil > alluvial soil > red soil (Figure 6).

### 3.2. Factors that Influence Soil Cd Enrichment

#### 3.2.1. Topography and the Quaternary Geological Conditions

The results showed that the distribution of Cd in topsoil had good corresponding relationships with the topography and quaternary geological conditions. The soil Cd in the sea–land plains and tidal flats landforms, and the quaternary strata for the lantern in the land–sea alluvial sand group ($Q_{4dl}^{3mc}$) areas, was significantly higher than for the low hills landform and the alluvial and eluvial soil ($Q^{el}$) regions. The Cd content also corresponded well to the changes in grain size of the core sediments. The Cd concentration was low in coarse sediment dominated by sandy soil, but high in fine sediment dominated by cohesive soil. The SOM and CEC were both relatively high in the fine-grained sediments dominated by silty clay, which indicates that marine sediments contributed to Cd enrichment [38].

#### 3.2.2. Stream Transport 

Rivers play an important role in receiving and transporting pollutants because they are bodies of water that accept pollutants from both point sources (industry and mining) and non-point sources (urban life, agriculture, and atmospheric precipitation). They are also the sources of large rivers and oceans [39]. Since the 1980s, the rapid economic development of the Pearl River Delta has meant that a large number of heavy metal pollutants have been discharged into the local rivers, and most of them have come from the point sources of Shaoguan, Foshan, Guangzhou, and other big cities with advanced metallurgical and chemical industries. Since 2000, several heavy metal pollution emergencies in tributaries of the Pearl River have been reported [40,41]. The maximum values for Cd were mostly distributed on both sides of the study area watercourse, and the drill core differences for total Cd were consistent in terms of the geographical location of the drilling in the horizontal plane. The Cd levels were high in NSGC05 and NSGC11, but were low in NSGC27 and NSGC39, which reflected the sediment quality restrictions on elemental distribution during stream transportation. In general, the NSGC05 and NSGC11 cores contained more sediment transported from the Xijiang and Beijiang rivers, whereas the NSGC27 and NSGC39 contained more sediment from the Dongjiang River. The areas with high Cd contents were mainly distributed in the Xijiang–Beijiang delta sedimentary area, whereas the Dongjiang and the Tanjiang delta sedimentary area had lower Cd contents [42,43]. Therefore, in addition to human activities, stream transport and delta deposition are also factors that have led to the high degree of soil Cd pollution in this study area.

#### 3.2.3. Physical and Chemical Properties of Soils

In the topsoil samples, the content of cadmium was weakly correlated with pH, SOM, and CEC, and the correlation coefficients were 0.25, 0.159, and 0.171, respectively (Table 4), due to different topographies and soil genetic type. However, the trend of Cd content in soil profile was significantly positively correlated with the trend of SOM and CEC, *P* < 0.001, which indicated that SOM and CEC were also important factors affecting the Cd distribution in the soil. In general, the main influencing factors on the formation of the existing soil Cd distribution pattern in the study area were topography, geological conditions, and stream transport, but soil type and soil physicochemical properties also affected Cd distribution.

#### 3.2.4. Human Engineering Activities

The background Cd concentration of the soil in Guangdong Province is 0.056 mg/kg. The “Risk Screening Values for soil heavy metal: The Pearl River Delta Area (DB44/T1415-2014)” showed that the background concentration for the soil environment in the Pearl River Delta had doubled to 0.11 mg/kg compared to Guangzhou, whereas the average value for soil Cd at 1.5 m to 2.0 m depths in Nansha was 0.42 mg/kg. This meant that it was three times higher than the background concentration of the soil in the Pearl River Delta, a difference most likely due to differing natural geological factors. The average Cd concentration in the topsoil (0.54 mg/kg) was higher than that at 1.5 m to 2.0 m depths. The overall spatial variability of the study area was moderate. The CV for paddy soil is 28%, whereas our soils were above 65% because they were more severely affected by human engineering activities. In addition, the variation coefficient for Cd content in the topsoil (0–20 cm) was larger than for the deep soil (20–200 cm). The Cd concentration for the different deep soils tended to be consistent at the same depth. The above results indicate that human inputs could have polluted the surface soil with Cd to some extent, and that the influence of human activities is random under natural geological conditions.

### 3.3. Evaluation of Soil Cd Pollution

The geo-accumulation index (Igeo) was used to assess pollution in the study area, and the Cd concentration of the Guangdong Province soil was selected as the geochemical background concentration for Cd [39] when attempting to describe the high concentrations of heavy metals in sediments. The median Igeo values of each element were in the order Cd > Cu > Zn > As > Cr > Hg > Pb from large to small (Figure 7), where the mean and median Igeo values for Pb, Hg, and Cr were all lower than 1. The Pb, Hg, and Cr levels indicated that the Nansha sediments were generally uncontaminated to moderately contaminated with these metals. However, discrete points appeared locally under the influence of point sources; for example, Pb levels achieved a heavily contaminated classification, Hg levels achieved a moderately to heavily contaminated classification, and the largest Igeo Cr value was 7.19, which was recorded in an extremely contaminated area of the abandoned Meishan industrial park, Huangge town, Nansha District. The pollution classification for As was mainly uncontaminated to moderately contaminated; the maximum Igeo was less than 2, and most local samples were uncontaminated. The pollution classification for Zn was mainly uncontaminated to moderately contaminated; the maximum Igeo was 3.56, but some local areas were heavily contaminated. The Igeo for Cu ranged from −1.61 to 2.41, which meant that the study area was mainly moderately contaminated, but some local areas were moderately to heavily contaminated. Cadmium had the highest Igeo of 4.99, the highest Igeo mean of 2.54, and the highest median of 2.68 among all tested metals. This means that Cd was the most serious polluter of the heavy metals in the study area. Furthermore, human activities had a considerable impact on the ground accumulation index. More than 65% of the Igeo was within the range of 2 to 3 (Figure 8), which indicated that the study area was mainly moderately to heavily contaminated. However, more than 17% was heavily contaminated, and some local areas were heavily to extremely contaminated. The variation coefficient for the Cd Igeo was the smallest among all tested metals (CV = 37.5%), which indicates that the Cd pollution in Nansha District was the most serious among all heavy metal elements, and the most common. In the Nansha District, the results showed that 52% of the area was considered to be at very high risk from Cd pollution. Furthermore, 36% for the study area was classified as dangerous, and these areas were mainly distributed in Hengli, Wanqingsha, Xinken, and other towns.

In general, human engineering activities have a great impact on Igeo and often result in moderate to heavy contamination. However, Cd pollution distribution is universal, and the potential ecological risk caused by Cd poses a considerable threat. The natural geological features of the Pearl River Delta and human activity have resulted in moderately to heavily Cd contaminated soil in Nansha. Therefore, it is necessary to determine the influencing factors and the extent of their effects on soil Cd pollution, and to strengthen production management and process controls to prevent continual increases in soil heavy metal contents. Planting structures in farmland also need to be adjusted in order to avoid heavy metal effects on human health via the food chain.

### 3.4. Chemical Fractionation of Soil Cd

#### 3.4.1. Cd Fractionation and Lability

It is generally agreed that the harmful effects of heavy metals on human beings and the environment depend not only on their total contents, but also on their chemical composition. After Cd enters the soil, most reacts with inorganic and organic components through adsorption, complexation, or precipitation to form carbonate-bound, humic-acid-bound, Fe–Mn-oxide-bound, organic matter-bound, and other forms. Only a few exist in water-soluble and exchangeable forms, which can effectively affect soil microbial metabolic activity. The water-soluble, exchangeable, and carbonate forms of Cd in the environment are sensitive to changes in the environment, which may affect activity and migration [44]. The Cd that is bonded to Fe–Mn oxides, organic matter, and residual forms is relatively stable in the environment and has weak activity. Therefore, the instability of Cd needs to be characterized using the sum of the water-soluble, ion-exchange, and carbonate forms, which is called labile Cd. The contribution rate of each component to the total concentration (Figure 9) indicated that in all the core sections, the proportion of the water-soluble form was relatively low, whereas the exchange form and carbonate form proportions were relatively high. The labile Cd proportion ranged from 37.92% to 47.49%, and averaged 44.19% of the total Cd.

#### 3.4.2. Bioavailability of Cd across the Area

The order of bioavailability of different heavy metal forms in soil by sequential extraction procedures was: water-soluble > exchangeable form > carbonate-bound > humic-acid-bound > Fe–Mn-oxide-bound > organic-matter-bound > residual form [45,46]. The fractions were classified into three bioavailable categories: easily bioavailable, moderately bioavailable, and inertly bioavailable [47]. Easily bioavailable Cd is the sum of water-soluble and exchangeable form, which has great mobility and is most easily absorbed by organisms. The moderately bioavailable Cd is the sum of the carbonate, humic acid, Fe–Mn oxide, and organic matter forms, which can be converted into bioavailable Cd under strong acid or other appropriate conditions. The inertly bioavailable or residual form has little effect on the migration and bioavailability of cadmium in soil. The bioavailability of Cd in the study area indicated that the bioavailable Cd in cores ranged from 0.004 to 0.37 mg/kg, with an average of 0.064 mg/kg, and moderately bioavailable ranged from 0.022 to 0.336 mg/kg, with an average of 0.126 mg/kg (Table 5). The bioavailable and moderately bioavailable Cd values were generally higher at sampling depths of less than 20 meters (Figure 10), and the average of the bioavailable and moderately bioavailable Cd was 0.077 and 0.156 mg/kg, respectively, while the values taken at sampling depths of more than 30 meters were generally lower, and the average of the bioavailable and moderately bioavailable Cd was 0.024 and 0.064 mg/kg, respectively.

#### 3.4.3. Factors Affecting Bioavailable Cd

Bioavailable Cd is considered to be a multi-equilibrium fraction between the solution and soil constituents [48], and positively correlated to the immobilized Cd in soil constituents. The majority of the soil Cd was bonded to the solid constituents and determined the total Cd in the studied soils. Therefore, there was usually a significant correlation between labile and total Cd [17]. The labile Cd largely depends on the total amount and source of Cd in soils. The difference in the fractionation between the four profiles corresponded to the difference in the total Cd. The fractionation characteristics of Cd in the NSGC05 and NSGC11 profiles were relatively similar, and there was little difference between the proportions of each form in the two profiles. The Cd fractionation characteristics in the NSGC27 and NSGC39 profiles were also similar, and the contribution rate of each form to the total content was relatively stable. This further reflected the Cd source differences between NSGC05/NSGC11 and NSGC27/NSGC39. Factor analysis of total, bioavailable, and labile Cd (Table 6) showed that the total Cd in the soil had a strong positive correlation with bioavailable Cd, and an extremely strong positive correlation with labile Cd. The correlation coefficients for total Cd and labile Cd were 0.893, 0.891, 0.898, and 0.977 for each profile, respectively, while the correlation coefficients for total Cd and bioavailable Cd were 0.780, 0.773, 0.679, and 0.787 for each profile, respectively.

The variation coefficient can reflect the fluctuation range of statistical data. The variation coefficient results for Cd fractionation in each core (Table 7) showed that the average level for the water-soluble variation coefficient was high, and its dispersion degree was greater. NSGC27 and NSGC11 were as high as 321.79% and 264.81%, respectively. However, the remaining morphological variation coefficients were relatively stable. Figure 8 also shows that both NSGC27 and NSGC11 had abnormal water-soluble points. The water-soluble Cd in NSGC27 at the depth of 40.7 m was 0.07 mg/kg, which accounted for 50.45% of the total Cd. The corresponding pH was 3.51. The water-soluble Cd in NSGC11 at depths of 16.2 m and 18.8 m were 0.188 and 0.074 mg/kg, respectively, which accounted for 33.38% and 26.54% of the total Cd, and the corresponding pH values were 4.03 and 4.21, respectively. The remaining samples were basically neutral to weakly alkaline, and the water-soluble contents were all less than 0.006 mg/kg. It is generally accepted that pH is one of the most important factors affecting Cd migration and transformation, but the negative correlation between the pH and the total Cd in the drilled samples was not significant because the pH values of the tested soil samples were generally neutral to weakly alkaline. In spite of this, the absolute values of the correlation coefficients between pH and the water-soluble Cd were greater than 0.6 (P < 0.01), and the squared correlation coefficients (R^2^) were 0.9833, 0.7029, 0.8422, and 0.8225 for the four profiles, respectively, which showed that there was a significant negative correlation between pH and water-soluble Cd. As pH increased, the water-soluble Cd decreased in an “inverted J” type manner, and when the pH exceeded 7, the water-soluble Cd approached the detection limit (Figure 11). The above results showed that the influence of pH on the migration and transformation of Cd was mainly to change the bioavailability of Cd, rather than the total Cd levels, which affected the ecological effectiveness of Cd.

## 4. Conclusions

The soil Cd concentration in Nansha District was significantly higher than the background concentration for Guangdong Province. It exceeded the risk screening value in the Soil Environmental Quality Risk Control Standard for Soil Contamination of Agricultural Land (GB15618-2018) of 0.3mg/kg. The topsoil Cd had good corresponding relationships with topography, quaternary geological conditions, and soil type. Namely, the Cd in the land–sea alluvial sand group ($Q_{4dl}^{3mc}$) areas was significantly higher than in the alluvial and eluvial soil ($Q^{el}$) regions, which corresponded to the grain size of the core sediments, and was ranked as coastal sand > swamp soil > paddy soil > fluvo-aquic soil > alluvial soil > red soil. Heavy metal pollution events frequently occur in tributaries of the Pearl River, and the rivers receive and transport pollutants. Therefore, river transport affects the distribution of Cd. Human engineering activities have had a considerable impact on the Igeo value and have resulted in moderate to heavy contamination, which has led to a potential ecological risk. The CV for Cd in topsoil (0–20 cm) was greater than for deep soil (20–200 cm), but the Cd levels tended to be the same as depth increased. There was a strong positive correlation between total Cd and bioavailable Cd in soil, and there was an extremely strong positive correlation with labile Cd, which accounted for about half of the total Cd. The influence of pH on the migration and transformation of Cd was mainly to change the bioavailable Cd rather than the total Cd levels, which affected the bioavailability of Cd. The bioavailable and moderately bioavailable Cd contents at sampling depths of less than 20 meters were generally higher than at the sampling depths of more than 30 meters. Furthermore, the change trend for Cd in soil profile was significantly positively correlated with soil organic matter (SOM) and cation exchange capacity (CEC), which indicates that the physical and chemical properties of the soil also affected the distribution of Cd. Effective treatment should be used to immobilize Cd before reclaiming these soils and prevent heavy metal increases in farmland, and the planting structure should be adjusted to avoid the impact of heavy metals on human health through ingestion via the food chain.

## Figures and Tables

**Figure 1 ijerph-16-03637-f001:**
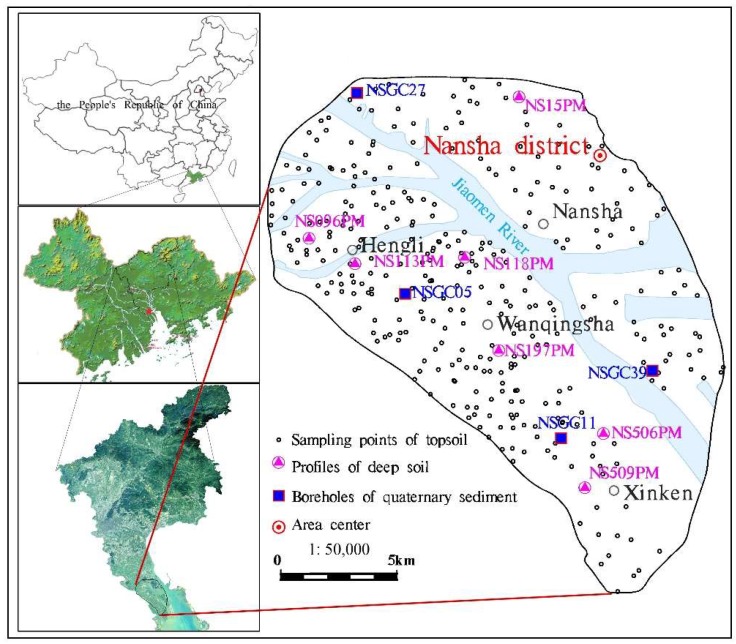
Topsoil sampling sites in Nansha District (Guangzhou, Guangdong Province, China).

**Figure 2 ijerph-16-03637-f002:**
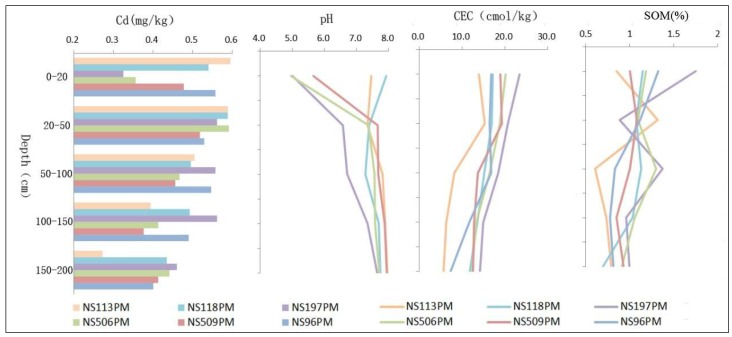
Comparisons between Cd, pH, cation exchange capacity (CEC), and soil organic matter (SOM) for the different depths and soil profiles.

**Figure 3 ijerph-16-03637-f003:**
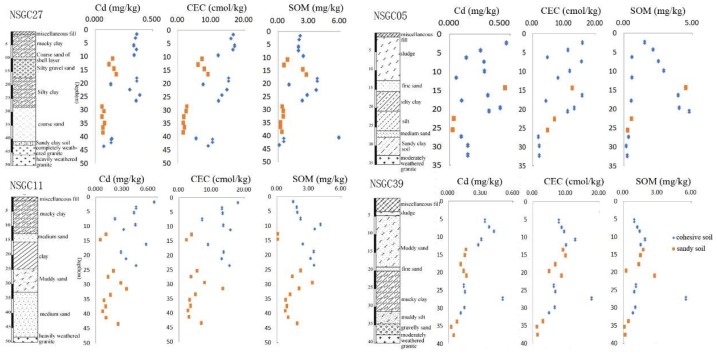
Cd, cation exchange capacity (CEC), and soil organic matter (SOM) distribution in the cores.

**Figure 4 ijerph-16-03637-f004:**
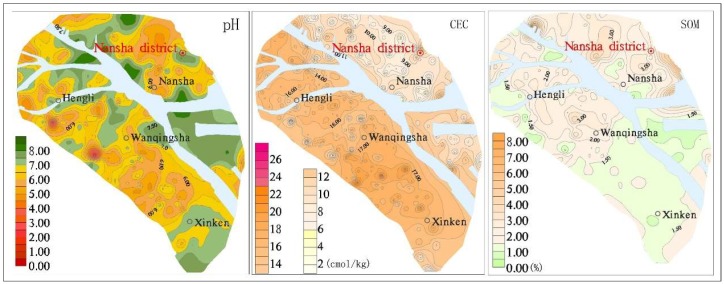
Contour map of pH, cation exchange capacity (CEC), and soil organic matter (SOM).

**Figure 5 ijerph-16-03637-f005:**
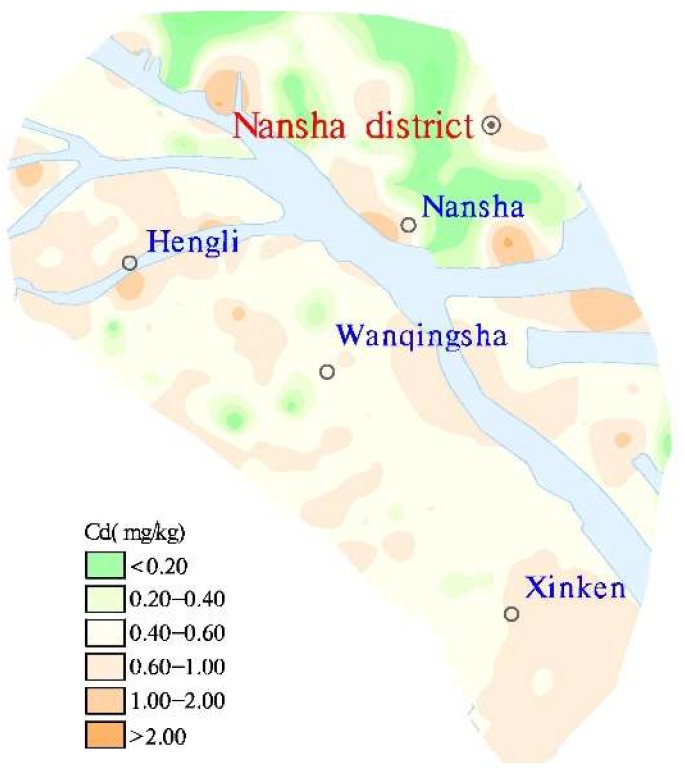
Soil Cd contour distribution.

**Figure 6 ijerph-16-03637-f006:**
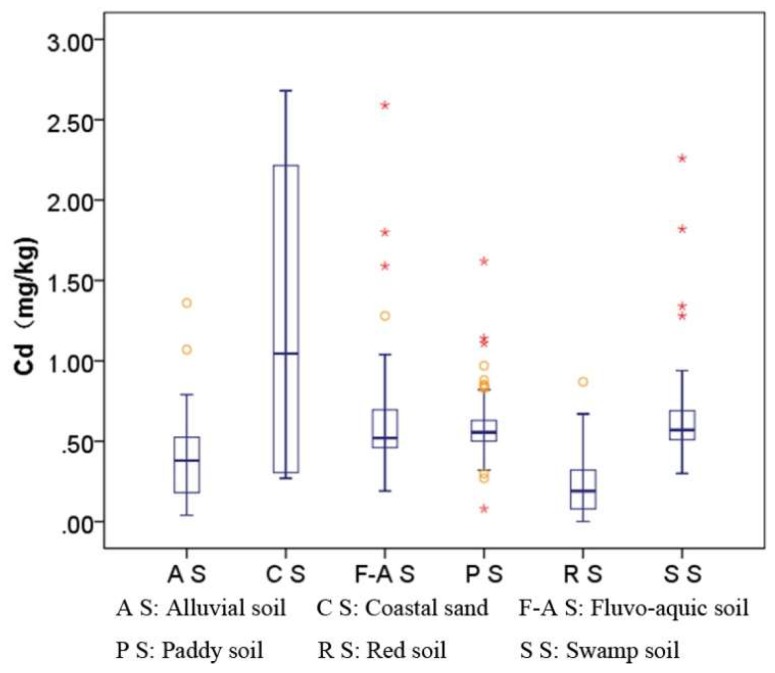
Box charts for Cd content of the different soil types.

**Figure 7 ijerph-16-03637-f007:**
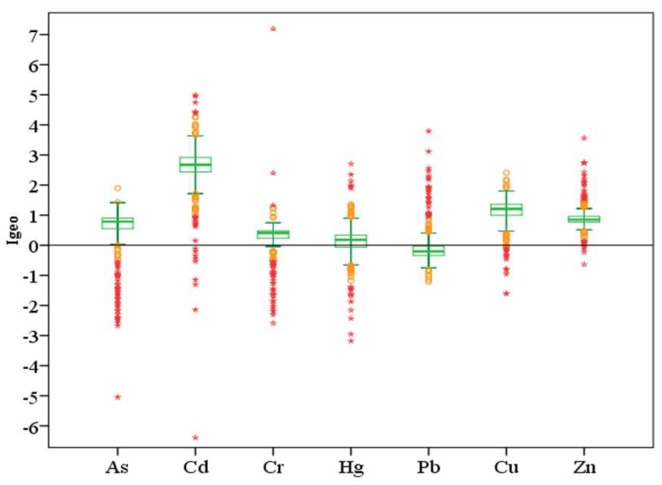
Igeo box plots for each element (N = 348).

**Figure 8 ijerph-16-03637-f008:**
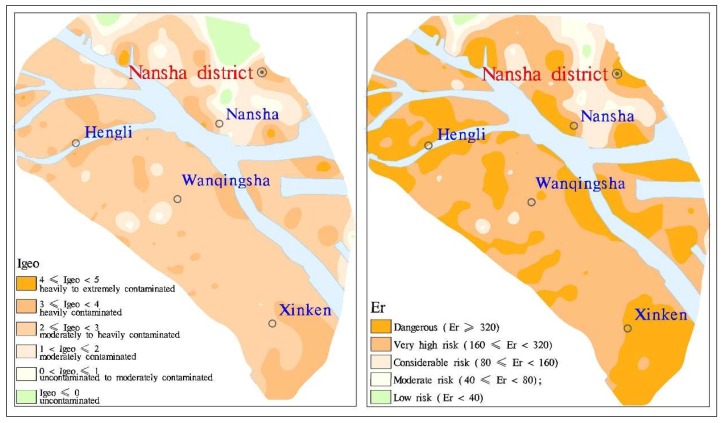
Soil Cd evaluation using the Geological Accumulation Index (**left**) and the Potential Ecological Risk Index (**right**) in Nansha District.

**Figure 9 ijerph-16-03637-f009:**
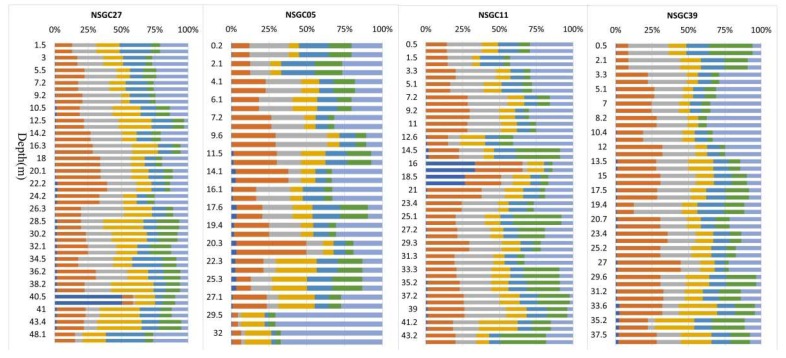
Percentage of Cd fractionation in each core.

**Figure 10 ijerph-16-03637-f010:**
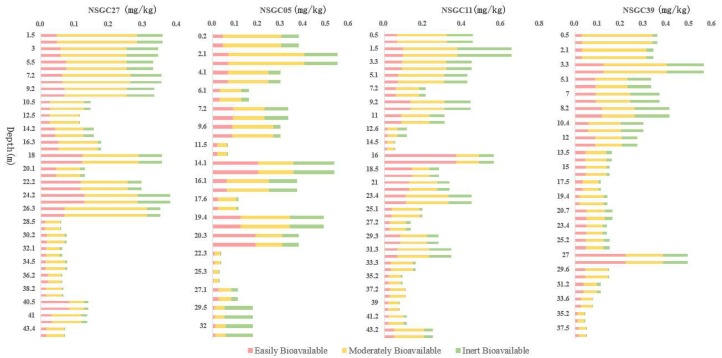
Bioavailability of Cd in each core.

**Figure 11 ijerph-16-03637-f011:**
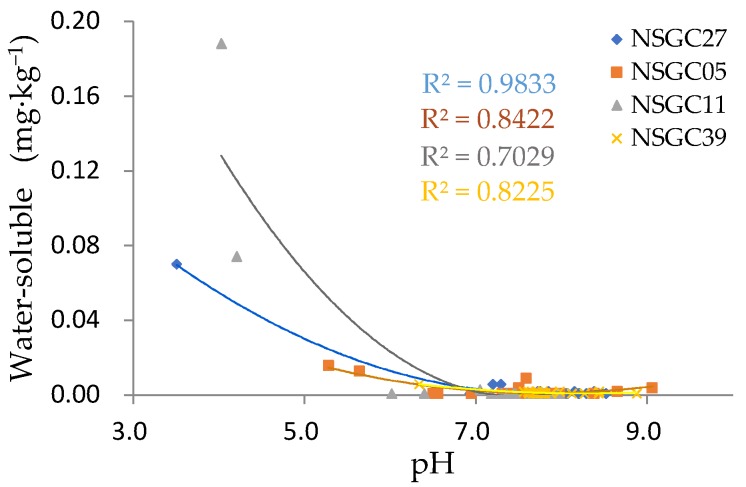
Correlation between Cd water solubility and the pH value for each core.

**Table 1 ijerph-16-03637-t001:** Sequential extraction methods for Cd in different depth soils from the study area.

Step	Fractions	Extract Composition	pH	Notes
F1	Water soluble	25 mL ultrapure water	pH = 8.0 ± 0.2	30 min ultrasound (40 KHz, 25 ± 5 °C), rinse with ultrapure H_2_O.
F2	Exchangeable	25 mL 1.0 mol/L MgCl_2_·6H_2_O	pH = 7.0 ± 0.2	30 min ultrasound (40 KHz, 25 ± 5 °C), rinse with ultrapure H_2_O.
F3	Carbonate-bound	25 mL 1.0 mol/L CH_3_COONa·3H_2_O	pH = 5.0 ± 0.2	60 min ultrasound (40 KHz, 25 ± 5 °C), rinse with ultrapure H_2_O.
F4	Humic-acid-bound	50 mL 0.1 mol/L Na_4_PO_7_·10H_2_O	pH = 10.0 ± 0.2	40 min ultrasound (40 KHz, 25 ± 5 °C), rinse with ultrapure H_2_O.
F5	Fe–Mn-oxide-bound	50 mL 0.25 mol/L HONH_3_Cl–HCl		1 h ultrasound (40 KHz, 25 ± 5 °C), rinse with ultrapure H_2_O.
F6	Refractory-organic-matter-bound	3 mL HNO_3_ + 5 mL 30% H_2_O_2_	pH = 2.0 ± 0.2	1.5 h bath (83 °C, stirred every 10 min), another 1.0 h bath with 3 mL 30% H_2_O_2_ (83 °C, stirred every 10 min), rinse with ultrapure H_2_O.
F7	Residual	5 mL mixture of 37% HCl, 70% HClO_4_, and 70% HNO3 (1:1:1)/5 mL 40% HF		Digested at 105 °C for 3 h.

**Table 2 ijerph-16-03637-t002:** Cadmium contents in the different cores.

Boring Number	Number of Samples	MIN ^1^	MAX ^2^	AVG ^3^	MID ^4^	SD ^5^	CV ^6^
		Unit: μg/Kg					
NSGC27	24	55	388	209	160	126	0.601
NSGC05	7	28	560	273	305	170	0.624
NSGC11	23	54	664	299	286	167	0.577
NSGC39	21	29	573	238	165	148	0.625

^1^ Minimum value, ^2^ Maximum value, ^3^ Average value, ^4^ Mid-value, ^5^ Standard deviation, ^6^ Coefficient of variation.

**Table 3 ijerph-16-03637-t003:** Descriptive statistics of pH, cation exchange capacity (CEC), and soil organic matter (SOM) in different soil types (n = 348).

	Coastal sand (n = 4)			Fluvo-aquic soil (n = 47)			Red soil (n = 27)		
	pH	CEC	SOM	pH	CEC	SOM	pH	CEC	SOM
MAX ^1^	8.30	5.52	4.05	8.28	19.65	12.17	8.43	13.81	6.87
MIN ^2^	6.88	3.13	0.20	4.60	2.83	0.43	4.05	4.10	0.53
AVG ^3^	7.59	4.51	1.82	6.76	14.29	2.46	5.89	7.06	2.62
MID ^4^	7.59	4.69	1.51	6.92	16.22	2.19	5.44	6.63	2.45
SD ^5^	0.64	0.87	1.60	1.04	4.71	1.62	1.19	2.35	1.38
CV ^6^	0.08	0.19	0.88	0.15	0.33	0.66	0.20	0.33	0.53
	Paddy soil (n = 158)			Alluvial soil (n = 32)			Swamp soil (n = 77)		
	pH	CEC	SOM	pH	CEC	SOM	pH	CEC	SOM
MAX ^1^	8.13	27.54	3.53	8.28	20.80	7.04	8.43	19.99	5.82
MIN ^2^	4.28	7.98	0.77	4.63	4.50	0.00	5.84	5.94	0.55
AVG ^3^	6.17	17.52	1.71	7.16	9.41	2.46	7.50	13.54	1.64
MID ^4^	6.15	17.43	1.48	7.40	8.24	2.50	7.56	14.06	1.52
SD ^5^	0.98	2.89	0.57	1.00	4.10	1.31	0.37	3.31	0.79
CV ^6^	0.16	0.17	0.33	0.14	0.44	0.53	0.05	0.24	0.49

^1^ Maximum value, ^2^ Minimum value, ^3^ Average value, ^4^ Mid-value, ^5^ Standard deviation, ^6^ Coefficient of variation.

**Table 4 ijerph-16-03637-t004:** Correlation and significance of cadmium concentration with cation exchange capacity (CEC), pH, and soil organic matter (SOM) in topsoil and four different cores.

		Topsoil	NSGC27	NSGC05	NSGC11	NSGC39
Correlation coefficient	CEC	0.159	0.912	0.760	0.805	0.820
	pH	0.250	−0.124	−0.177	−0.155	−0.333
	SOM	0.171	0.503	0.741	0.499	0.552
Sig (single side)	CEC	0.001	0.000	0.000	0.000	0.000
	pH	0.000	0.200	0.159	0.152	0.016
	SOM	0.000	0.000	0.000	0.000	0.000

**Table 5 ijerph-16-03637-t005:** Descriptive statistics of bioavailable and moderately bioavailable Cd in each core.

Boring Number	Bioavailable Cd				Moderately Bioavailable Cd			
	NSGC27	NSGC05	NSGC11	NSGC39	NSGC27	NSGC05	NSGC11	NSGC39
MAX	0.127	0.202	0.370	0.220	0.242	0.336	0.277	0.308
MIN	0.012	0.004	0.013	0.010	0.039	0.022	0.038	0.029
AVG	0.051	0.062	0.077	0.060	0.117	0.128	0.137	0.123
0–20 m AVG	0.056	0.084	0.104	0.063	0.146	0.165	0.157	0.154
>30 m AVG	0.029	0.010	0.035	0.022	0.069	0.045	0.100	0.043

**Table 6 ijerph-16-03637-t006:** Correlation coefficients for total, bioavailable, and labile Cd in each profile.

Fractions	NSGC27			NSGC05			NSGC11			NSGC39		
	Total	Bioavailable	Labile	Total	Bioavailable	Labile	Total	Bioavailable	Labile	Total	Bioavailable	Labile
Total	1.00	0.780	0.893	1.00	0.773	0.891	1.00	0.679	0.898	1.00	0.787	0.977
Bioavailable	0.780	1.00	0.916	0.773	1.00	0.920	0.679	1.00	0.908	0.787	1.00	0.885
Labile	0.893	0.916	1.00	0.891	0.920	1.00	0.898	0.908	1.00	0.977	0.885	1.00
Water-soluble	−0.064	0.266	0.030	0.555	0.876	0.696	0.345	0.877	0.635	0.390	0.674	0.478
Exchangeable	0.827	0.917	0.928	0.782	0.999	0.928	0.841	0.892	0.962	0.790	1.00	0.887
Carbonate	0.846	0.653	0.902	0.774	0.478	0.783	0.749	0.121	0.526	0.895	0.457	0.819
Humic acid	0.756	0.397	0.534	0.843	0.640	0.671	0.831	0.747	0.869	0.843	0.592	0.809
Fe–Mn oxide	0.773	0.314	0.430	0.778	0.331	0.541	0.795	0.170	0.484	0.674	0.243	0.582
Organic	0.796	0.527	0.620	0.622	0.174	0.364	−0.004	−0.202	−0.164	0.388	−0.084	0.233
Residual	0.932	0.655	0.746	0.800	0.539	0.558	0.893	0.339	0.633	0.868	0.777	0.869

**Table 7 ijerph-16-03637-t007:** % Total and variation coefficients for Cd fractionation in each core.

Fractionation	% Total (%)				Variation Coefficients (%)			
	NSGC27	NSGC05	NSGC11	NSGC39	NSGC27	NSGC05	NSGC11	NSGC39
Water-soluble	3.07	1.44	3.15	0.89	321.79	83.19	264.81	70.9
Exchangeable	22.13	20.31	22.69	25.65	31.5	51.47	25.69	32.81
Carbonate	20.43	16.17	19.86	20.95	35.29	53.01	32.56	33.04
Humic acid	18.06	13.37	11.56	12.93	42.07	47.76	35.56	44.02
Fe–Mn oxide	13.01	11.86	10.22	10.82	42.07	45.07	29.66	33.37
Organic	8.83	10.2	12.3	11.39	43.99	57.61	79.45	59.94
Residual	14.46	26.66	20.22	17.36	53.79	65.73	53.56	59.92

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
