# Peer review of "Distribution, Ecological Risk Assessment, and Bioavailability of Cadmium in Soil from Nansha, Pearl River Delta, China"

_ijerph, 2019, doi:10.3390/ijerph16193637_

Round 1

Reviewer 1 Report

The manuscript has investigated Cd contamination in a specific area of China. The manuscript notes that a range of metals have been determined, but only Cd focussed on.  The authors have selected a few samples to perform basic soil analysis on, such as sequential extraction procedures. I find the manuscript to be routine in nature and applied to just one area of china. I fail to see what interest an international reader could have on this paper. The use of ratio of metal content to background is very basic. This ratio multiplied by a constant factor I find counter productive. This could be acceptable in the distant past, but not  in the present day. At the very least, some attempt on defining bioavailability of Cd across the area is needed.

Reviewer 2 Report

In this study, the authors evaluated the degree of contamination, the bioavailability and the ecological risk of Cd in the region of Pearl River delta, China. Given the deleterious effects to humans associated to Cd exposure, mainly through food chain, the results of this research are of great interest, also in view of reducing and controlling pollution sources and in order to implement remediation activities of highly contaminated soils.

My comments are listed in order of appearance.

Line 132. Consider to delete “and”.

Line 196. “to the” has been repeated twice.

Lines 191-193. The units of measure appear wrong if they refer to pH and cation exchange capacity. In addition, the values reported appear different from those in Figure 4.

Line 197. Consider to write here the acronym for “coefficient of variation”.

Lines 203-204. Generally, the cation exchange capacity decreases with decreasing pH. Could the authors explain this statement? The same concept has been stated at lines 196-197.

Lines 209-210. µg/kg or g/kg?

Figure 5. Consider including full names of the abbreviations mentioned (AVG, SD, CV, CEC, TOC). Moreover, it is unclear whether the values of Cd reported for Chinese Soil, Guangdong Soil, Continental Crust Abundance and Pearl River Sediment are minimum or average.

Line 263. It is unclear the meaning of “negatively correlated with pH change”. In the sense that Cd levels increase as pH decreases?

Lines 278-281. Consider to include a brief description of CV, repeatedly mentioned in the text, in Methods section.

Lines 295-297. These two sentences seem to contradict each other. Probably, the second sentence should be changed in: “Pb levels CAN ACHIEVE a heavily contaminated classification”.

Line 299. The largest Igeo Cr value of 7.19 was not reported in the graph. Indeed, at line 306, the authors state that “Cadmium had the highest Igeo of 4.99”, which is properly reported in Figure 7.

Lines 368-369. This sentence appears in contrast with results reported in Table 2. In particular, there is a negative correlation coefficient between water soluble Cd and total Cd in NSGC27 profile.

Lines 407-410. This paragraph is not clear. Consider to rephrase it.

Line 420. Consider to replace “etc.” with “other towns”.

Reviewer 3 Report

The manuscript reports data on Cd distribution on soils from the Nansha region in China. The abstract should be better structured, with clear mention of the type and number of samples studied and the analyses performed. The introduction is well organized, with the objectives of the work clearly stated. An adequate amount of samples were used in the study, and the methodology section is well explained, although more detail is needed at some parts (please see comments below). However, the results are exposed in a mostly confusing manner. In this sense, I would suggest to slightly reorganize the section, reporting in separate sections all the data from soil and sediment profiles and then data from the surface samples. Conclusions present the same organization problems as the results section and should be rewritten incorporating the main findings of the whole study; in their current form they only include very general statements from some parts of the work. Overall, the manuscript needs revision before being considered for publication, in particular in the results and conclusions sections.

Specific comments:

Abstract: clearly state the number of soils analyzed in the study. Also, please explain which sediments were investigated: river sediments?

Line 44: 7% of what? Surface?

Line 46 What is “Cadmium rice”?

Line 96 Add the number of deep profiles that were sampled

Line 101: To which depth were sediments sampled?

Line 112 and following: please explain in more detail the methodology for the fractionation of Cd
Line 165 Why are CEC, pH etc… for the surface soils not reported? They should be before explaining Cd contents

Line 169 Can you add the reference values for Cd pollution?

Line 190 and Figure 4 Data for the soil profiles should be presented using similar plots as the sediments (Figure 5)

Table 1 needs to be remade. Neither the properties nor the units are presented in the table, so it is impossible to know what the numbers represent.

Lines 258-260 What do you mean “when the topography and soil genetic type were compared”? Also, please show correlations (and their statistical significance) in a Table, it would be easier to read.

Line 293 and Figure 6: why are results from other elements than Cd reported?

Section 3.4 Please change the titles and subtitles in this section, that reports results from a chemical fractionation of Cd, not bioavailability or morphology or activity.

Line 347 and following Please replace “state” with “fraction” or “form”

Line 356 and following: Please use “fractionation” instead of “morphological distribution” and “morphology”.

Section 3.4.2 How were dissolved and labile Cd determined?

Table 2 Why were correlations calculated for each core and not for the whole dataset?

Table 3 Please revise units, are you sure that mean vales are given in %?

Lines 401-412: These whole paragraph should be removed, as it is unclear if it is results or discussion and is poorly written.

Line 401 What does “activities of the heavy metals” mean here? Only Cd is reported

Line 404 Where are the results of the stepwise regression analyses shown?

Line 408 What does “organic state” mean?

Please add results from Cd deep distribution to the conclusions.

Line 417 Please do not mention other metals in the conclusions, otherwise you should include a section on the paper with the spatial distribution of all the other elements in the soils

Line 420 Please remove the sentence “it is mainly distributed in Hengli…”

Lines 421-423 Please briefly explain in which way does each of these factors affect Cd concentrations

Line 425 What is Cd activity?

Line 427 What are the “reactive fractions”? Please use scientific terms properly.

Round 2

Reviewer 3 Report

The manuscript has been improved by accomplishing several changes in structure and adding necessary explanations. Only minor changes are necessary, as well as a deep revision of grammar by an English native speaker.

Lines 132-134 I think that the details about the extraction scheme could be included in the paper and not only in the supplementary material

Line 284 Please rewrite this sentence: “In the topsoil samples, cadmium…”

Table 6 Please write “% total” instead of “percentage mean”

Line 447 Please replace “by 0.3” with “of 0.3”

Author Response

Dear Reviewer:

We are truly grateful to yours and other reviewers’ critical comments and thoughtful suggestions on our manuscript entitled “Distribution, Ecological Risk Assessment and Bioavailability of Cadmium in Soil in Nansha, Pearl River Delta of China”. (ID: ijerph-585448). Based on these comments and suggestions, we have made careful modifications on the original manuscript. All changes made to the text are in yellow color. In addition, we have consulted native English speakers for paper revision before the submission this time. We hope the new manuscript will meet your magazine’s standard. Below you will find our point-by-point responses to the reviewers’ comments/ questions:

Point 1: Lines 132-134 I think that the details about the extraction scheme could be included in the paper and not only in the supplementary material

Response 1: Table 1, line135. Considering the Reviewer’s suggestion, details about the extraction scheme have been added in the paper.

Point 2: Line 284 Please rewrite this sentence: “In the topsoil samples, cadmium…”

Response 2: Line 285. We have re-written this part according to the Reviewer’s suggestion, that is “In the topsoil samples, the content of cadmium was weakly correlated with pH, SOM and CEC, and the correlation coefficients were 0.25, 0.159 and 0.171 respectively”.

Point 3: Table 6 Please write “% total” instead of “percentage mean”

Response 3: Table 7. As Reviewer suggested that “percentage mean” has been rewritten as “% Total-average”.

Point 4: Line 447 Please replace “by 0.3” with “of 0.3”

Response 4: Line 445. We are very sorry for our incorrect writing “by 0.3”, it has been replaced with “of 0.3”.

Special thanks to you for your good comments. Looking forward to hearing from you.

Thank you and best regards.

Yours sincerely,

Authors:

Name: Fangting Wang

E-Mail: [email protected]

Name: Changsheng Huang *

E-Mail: [email protected]

Name: Zhihua Chen

E-Mail: [email protected]

Name: Ke Bao

E-Mail: [email protected]